# Confronting Secondary Metabolites with Water Uptake and Transport in Plants under Abiotic Stress

**DOI:** 10.3390/ijms24032826

**Published:** 2023-02-01

**Authors:** Juan Nicolas-Espinosa, Paula Garcia-Ibañez, Alvaro Lopez-Zaplana, Lucia Yepes-Molina, Lorena Albaladejo-Marico, Micaela Carvajal

**Affiliations:** Aquaporins Group, Plant Nutrition Department, Centro de Edafología y Biología Aplicada del Segura (CE-BAS, CSIC), Campus Universitario de Espinardo, Edificio 25, 30100 Murcia, Spain

**Keywords:** phenolic compounds, glucosinolates, plant secondary metabolites, salinity, drought, mineral nutrition, exudates, molecular transporters, aquaporins, water transport

## Abstract

Phenolic compounds and glucosinolates are secondary plant metabolites that play fundamental roles in plant resistance to abiotic stress. These compounds have been found to increase in stress situations related to plant adaptive capacity. This review assesses the functions of phenolic compounds and glucosinolates in plant interactions involving abiotic stresses such as drought, salinity, high temperature, metals toxicity, and mineral deficiency or excess. Furthermore, their relation with water uptake and transport mediated through aquaporins is reviewed. In this way, the increases of phenolic compounds and glucosinolate synthesis have been related to primary responses to abiotic stress and induction of resistance. Thus, their metabolic pathways, root exudation, and external application are related to internal cell and tissue movement, with a lack of information in this latter aspect.

## 1. Introduction

Plants must maintain their physiological state steady for normal growth. For this, the complex and interconnected secondary metabolic pathways must respond to different environmental changes [1]. The secondary metabolism is usually associated with the response of plants to abiotic stresses (heat, cold, salinity, drought, etc.), and metabolites mainly include phenols, terpenes, and nitrogen/sulfur-containing compounds. The increase of these metabolites allows plants to maintain their growth, yield, and reproduction capacities [2].

Previous studies have shown that moderate or controlled stress conditions could have a positive effect on the quality of several crops due to changes in secondary metabolism [3], and the routes that determine this beneficial effect have received considerable attention in the past few years. Thus, water stress, salinity, and nutrient stress have been described as stresses that could improve the quality of vegetables and fruits through the stimulation of the secondary metabolism pathways that synthesize highly bioactive compounds such as vitamins, pigments, phenolic compound, glucosinolates, etc. [4,5]. Furthermore, the relationship of secondary metabolism (mainly phenolic compounds and glucosinolates) with primary responses such as water uptake has been associated with abiotic stress tolerance [6]. This has prompted exploration of the external application of phenolic compounds and glucosinolates (GSLs), given their potential role as biostimulants that could solve detrimental effects of abiotic stress [7].

In this context, the following review provides a summary of the most up-to-date published information on this topic. It highlights the areas that need to be further investigated with respect to the involvement of phenolic compounds and GSLs in the responses of plants to abiotic stress and the associated mechanisms of action. The focus is placed on the fact that there is very little information in regard to the mechanism that mediates regulation of phenolic compounds and glucosinolates in certain physiological processes such as water uptake and transport. However, the results to date indicate that this evaluation should take into account the regulation of aquaporins.

## 2. Secondary Metabolism of Phenolics and Glucosinolates

### 2.1. Phenolic Compounds Biosynthesis

Plant phenolic compounds are plant secondary metabolites that can be divided into three main types: (I) polyphenols, including lignin and tannins, (II) oligophenols, such as flavonoids, stilbenes, and coumarins, and (III) phenolic acids, mainly including derivatives of benzoic acid and cinnamic acids (Figure 1).

The basic structure of phenolic acids is composed of one aromatic ring with one or more hydroxyl substituents, and they can mainly be divided into benzoic and hydroxycinnamic acids [8]. They are known to be involved in diverse functions including plant growth, defense, and protection against UV stress. Hydroxycinnamic acids are derived from cinnamic acid, and are often found as simple esters combined with quinic acid or glucose. In plants, the most common hydroxycinnamic acids are ferulic, caffeic, *p*-coumaric and sinapic acids, which possess a C6-C1 basic structure derived from cinnamic acid [9]. Meanwhile, hydroxybenzoic acids derive from benzoic acid, and include salicylic acid, vanillic and gallic acid, among others.

Their biosynthesis starts with the shikimate pathway, which converts carbohydrates from glycolysis and the phosphate pentose pathway into chorismate, which is the precursor of phenylalanine and tryptophan [10]. However, intermediate metabolites in this pathway are also converted into phenolic acids in a multistep process. Dehydroxyshikimic acid is produced on the third step of this pathway, and its conversion into gallic acid has been described, although the enzymes involved remain unconfirmed. Then, the shikimate produced after the action of the bifunctional enzyme 3-dehydroquinate dehydratase/shikimate dehydrogenase can be turn into quinic acid after dehydration by the enzyme quinate dehydratase. The enzyme hydroxycinnamoyl-CoA quinate:hydroxycinnamoyl transferase (HQT) then esterifies the quinic acid with caffeoyl-CoA, to produce chlorogenic acid.

Phenylalanine is the precursor of the phenylpropanoid pathway, which begins with the deamination reaction carried out by the phenylalanine ammonia-lyase (PAL) enzyme to form *trans*-cinnamic acid. This is a key regulatory step for the formation of many phenolic compounds. After that, the C4 position of the aromatic ring in the *trans*-cinnamic acid is hydroxylated by the 4-hydroxylase (C4H) enzyme, resulting in *p*-coumaric acid. C4H also interacts in a multi-enzyme complex with the *p*-coumaroyl (shikimate/quinate) 3-hydrolase (C30H) enzyme, which hydroxylates the *p*-coumaroyl residue or the free *p*-coumaric acid at C3, resulting in a residue of caffeoyl or a caffeic acid [11].

Lignin biosynthesis involves a sequence of hydroxylations and methylations of the aromatic ring, and side-chain modification of tyrosine and phenylalanine through the phenylpropanoid pathway [12]. This yields phenolic derivatives such as cinnamate, *p*-coumarate, caffeate, feruloate, and sinapate, with the production of this heteropolymer conferring elevated mechanical resistance to the cell wall [11].

Diverse sub-types of flavonoids can be found, including flavones, flavonols, isoflavonoids, and anthocyanins, among others. They are formed in further steps of the phenylpropanoid pathway, starting with *p*-coumaroyl CoA transformation into naringenin chalcone by the action of the chalcone synthase enzyme [13]. Then, chalcone isomerase turns naringenin chalcone into flavanones (such as naringenin or hesperidin). Afterwards, flavones can be produced due to the action of the flavone synthase enzyme. Two enzymatic steps carried out by the flavanone-3-hydroxylase and the flavonol synthase enzymes must take place to obtain flavanols from flavanones. Moreover, anthocyanins can be obtained from flavanones after four enzymatic steps, in which anthocyanidins and proanthocyanidins are also involved.

Salicylic acid can be biosynthesized from the isochorismate created in the shikimate pathway, or from the cinnamic acid from the phenylpropanoid pathway. However, although biochemical evidence of the production of salicylic acid from isochorismate has been described, the exact mechanism of this pathway remains unknown [14].

### 2.2. Physiological Role of Phenolic Compounds

Phenolic acids have a wide range of relevant biological roles, with some of them (such as hydroxycinnamic acids) involved in the biosynthesis of structural components of the cell wall [15]. Furthermore, their release into the environment through root exudation or by leaching from roots, aerial parts, or decomposing residues has been demonstrated to produce allelopathies in other plant crops [16]. In terms of their involvement against plant abiotic stresses, the accumulation of phenolic compounds has been reported as a response to the oxidative stress derived from heavy metal toxicity [17]. For example, in *Brassica juncea,* lead (Pb) stress increased total accumulation of polyphenol, flavonoids, and anthocyanins [18]. Water stress in cherry tomato has been demonstrated to regulate the phenylpropanoid biosynthetic pathway, increasing quercetin and kaempferol content, and protecting the plant against the H_2_O_2_ molecules produced in the cytoplasm [19]. In addition, the role of flavonoids against nutrient deficiency has been reported, where they are involved in mycorrhizal modulation [20] and thus promoting the nitrogen and phosphorus supply.

However, a deeper understanding of the primary molecular targets of these phenolic acids is required in order to clarify their role under abiotic stress. Thus, they have been reported to be involved in cell structural roles, connecting cell-wall polysaccharides and anchoring lignin to the polysaccharide domain [21]. Polyphenols are the result of the union of diverse phenolic acids, leading to polymers of high molecular weight (500 and 3000 Da) [22]. Molecules such as tannins and lignins are included in this group. Two different sub-groups of tannins can be found in plants: proanthocyanidins and gallo- and ellagi-tannins. These compounds are well known to be involved in redox control in the plant cell, providing UV protection against heat stresses and scavenging the damage provoked by free radicals [23]. Meanwhile, lignin is a polyphenol involved in plant growth and development that has been reported to enhance cell-wall rigidity, conferring hydrophobic properties that relate to water and mineral transport through the vascular bundles [24].

One of the main physiological roles of flavonoids is to cope with the abiotic stresses associated with the plant’s sessile lifestyle, such dealing with salt and drought stresses. Under these circumstances, specific ion reactive species are produced, such as peroxide ions, singlet oxygen, and superoxide, provoking oxidative damage in cells [25]. The main function of flavonoids is to act as antioxidant agents by scavenging the reactive species produced by different abiotic stresses. For example, studies on wheat showed increased biosynthesis and enhanced accumulation of total anthocyanin and phenolic content under drought stress [26].

Meanwhile, salicylic acid is well known as one of the main molecules involved in diverse plant signaling processes. This plant hormone is able to interact with others, in a positive or negative feedback loop, playing a part as a messenger in systemic acquired resistance and abiotic stress tolerance [27]. Salicylic acid has been reported to be a relevant regulator for sodium influx and efflux. Experiments performed with saffron under saline conditions have demonstrated that the foliar application of salicylic acid reduced osmotic stress and increased the K^+^/Na^+^ ratio of shoots [28].

### 2.3. Glucosinolate Biosynthesis

GSLs are sulfur-based molecules generated during secondary metabolism, mainly present in plants from the family Brassicaceae. They are B-thioglucosides-*N*-hydroxysulfates, with a side chain that varies depending on the amino acid from which they were derived. They are generally classified into three different groups: (i) aliphatics, derived from methionine, alanine, leucine, isoleucine and valine; (ii) indole, whose precursor is tryptophan; and (iii) aromatics, derived from phenylalanine and tyrosine [29].

With respect to GSLs biosynthesis, the identification of genes and enzymes involved in its pathway has been mostly deciphered in the plant model *Arabidopsis thaliana* by use of techniques such as mutation, knockouts, and cloning, or the identifitcation of quantitative trait locus (QTLs). The process can be described according to three differentiated phases: elongation of the precursor amino acid side chain, core structure formation, and side-chain modifications [30].

#### 2.3.1. Side Chain Elongation of the Precursor Amino Acid

This first stage takes place with the precursor amino acids methionine or phenylalanine. It starts with a deamination carried by the amino acid transferase enzyme BCAT, which provides a 2-oxoacid. Then, the 2-oxoacid is subjected to a cycle of three subsequent reactions of condensation, isomerization, and decarboxylation, carried out by MAM, IPMI, and IPM-DH enzymes, respectively (Figure 2). The result is a 2-oxoacid elongated in a methylene group, which could enter a new elongation cycle or be transaminated by BCAT to proceed to the structural formation of the glucosinolate core [31].

#### 2.3.2. Glucosinolate Core Structure Formation

About 13 enzymes have been reported to be involved in this step, which can be divided into five sub-steps. First, precursor amino acids are converted into aldoximes by the intervention of cytochromes P450 from the CYP79 family. It has been determined that CYP79B2 and CYP79B3 metabolize tryptophan, while CYP79A2 is associated with the pathway of methionine and its elongated derivatives. The resulting aldoximes are oxidized by cytochromes P450 from the CYP83 family, giving rise to diverse bioactive compounds. CYP83B1 has been demonstrated to be involved in the degradation of tryptophan and phenylalanine-derived acetaldoximes, and CYP83A1 is known to be involved in the oxidation of aliphatic-related aldoximes. Then, a conjugation by non-enzymatic reactions occurs between an activated aldoxime and a sulfur donor. The resulting *S*-alkyl-thiohydroximates are converted into thiohydroximates by the action of the *C-S* lyase SUR1 (Figure 2). These molecules are *S*-glycosylated by glycosyltransferases from the UGT74 family, with UGT74B1 acting on the derived compounds of phenylalanine, and UGT74C1 on the aliphatic pathway. As a consequence, the resulting desulfo-GSLs are sulfated by sulfotransferases. In *A. thaliana*, these enzymes are named SOT16, SOT17, and SOT18, but in *Brassica oleracea*, these are known as ST5a, ST5b and ST5c, respectively [31].

#### 2.3.3. Secondary Modification of the Glucosinolates Side Chain

The third stage of GSLs biosynthesis is a crucial point for increasing the biodiversity of GSLs in the different *Brassicaceae* species. In this context, modifications can be classified based on the type of GSL. Secondary modifications in aliphatic GSLs include oxygenations, hydroxylations, alkylations, and benzoylations. Furthermore, diverse loci related with these reactions have been identified, such as GS-OX, GS-AOP, and GS-OH. In addition, indole GSL’s secondary modifications encompass hydroxylation and methoxylation reactions. Between the enzymes involved in the hydroxylation reactions, cytochrome CYP81F is well known for its role in the glucobrassicin process [30].

### 2.4. Glucosinolates’ Physiological Role

GSLs are known to be involved in the responses of plants to environmental stresses. Despite the fact that intact GSLs might act against stresses, their mayor bioactivity relies on their degradation products, after the action of the β-thioglucosidase enzyme myrosinase [32]. Thus, GSLs and myrosinase together represent a two-component defence system. After tissue rupture as a result of cell damage, myrosinase comes into contact with GSLs, activating the production of a thiohydroximate-O-sulfate intermediate. Then, after desulfation, different reactions take place, providing diverse bioactive phytochemicals, including nitriles, epithionitriles, thiocyanates, nitriles, thiocyanates, oxazolidine-2-thiones, and isothiocyanates. Other proteins also intervene in this process, such as myrosinase-binding proteins (MBPs), myrosinase-associated proteins (MyAPs), and specifier proteins [33].

According to their role in abiotic stress response, it has been reported that myrosinase and GSLs are both present in stomatal guard cells, and recent studies have revealed changes in GSL metabolism in these cells after treatment with CO_2_ [34]. Moreover, stomatal closure has been observed after exogenous application of different GSL hydrolysis products. However, the mechanisms behind this defense system in vivo need to be better explained [35]. The accumulation of GSLs has also been linked to stresses such as drought and salinity [36]. In Chinese cabbage (*Brassica rapa* L. spp. *pekinensis*), an elevated accumulation of GSLs under drought conditions has been observed [37]. This physiological reaction has been correlated with stomatal closure, raising the hypothesis that an increase in GSLs directly or indirectly regulates stomatal movement in order to prevent water loss and improve the plant’s survival under these conditions [38]. With respect to salinity, studies performed in broccoli (*Brassica oleracea* L. var. *italica*) have demonstrated that GSL accumulation in inflorescences and leaves relating to higher tolerance was highly determined by the cultivar and the season [39].

## 3. Role of Phenolic Compounds and Glucosinolates under Abiotic Stress in Relation to Water Uptake and Transport

As a consequence of climate change, plants suffer extreme environmental conditions, which can reduce their growth and affect agricultural productivity. Currently, abiotic stresses including salinity, drought and high temperatures, mainly affecting water uptake and transport in the plant, are responsible for major losses in vegetable crops worldwide [40]. Human activity is increasing the concentration of greenhouse gases in the atmosphere, resulting in an increase in the earth’s temperature. Furthermore, some regions in the world are experiencing a reduction in the frequency and intensity of precipitation, causing an increase in drought events, high temperatures, nutrient toxicity, and salinity [41]. As a consequence of these extreme conditions, available soil water and relative air humidity are decreasing, leading to reduced plant growth, reduced metabolic demands, and ultimately, the premature death of plants [42]. Plants are able to detect these external environmental stresses, and can adapt and generate adequate responses to them [43]. It is known that secondary metabolites play a fundamental role in this adaptive capacity, allowing plants to survive in the different environmental conditions they experience [44].

### 3.1. Salinity

It is known that salinity reduces plant growth by different mechanisms which alter the water relations within the plant [45]. One relevant cause of this disturbance is the excess of Na^+^ and Cl^−^ in the surroundings reducing the osmotic potential of the soil solution and consequently water uptake by the plant root, causing osmotic stress in the plant. The maintenance of cell homeostasis under salinity has been related to reactive oxygen species (ROS) [46]. Phenolic compounds have been reported to participate in the deactivation of ROS, and their concomitant contribution to water and nutrient status has been revealed [44]. However, the direct relation has been poorly studied.

Several studies have shown how phenolic content varies under conditions of high salinity; however, the antioxidant response of plants to this stress strongly depends on the dose and timing of the salinity treatment, as well as the saline source [46]. It has been reported that the application of high salt concentrations in *Solanum lycopersicum* cv. Microtom resulted in elevated concentrations of phenolic compounds in leaves, which increased according to the salinity levels and the time of application [47]. Other researchers [48] reported different responses to salinity depending on the brassica subspecies involved. They observed that kale (*Brassica oleracea* var. acephala) and white cabbage (*Brassica oleracea* var. capitata) increased their total levels of phenolic acids, especially total hydroxycinnamic acids, which resulted in lower metabolic stress and consequently higher tolerance to salinity, while salt-sensitive Chinese cabbage (*Brassica rapa* ssp. pekinensis) showed a decrease in its phenolic acid levels, particularly protocatechuic, caffeic, salicylic, and 4-coumaric acids. Furthermore, a possible relationship between oleuropein and glucose was reported in *Olea europaea* L. It is suggested that oleuopein, in addition to acting as an antioxidant, could act as a glucose reservoir for osmoregulation to enable plants’ adaptation to salinity, due to the fact that this phenol is formed by a glucose molecule that can easily be cleaved by the endogenous or exogenous enzyme *β*-glucosidase [49].

It should be noted that in addition to containing phenols, the different species belonging to the Brassicales order have high contents of GSLs. These secondary metabolites are responsible for their odor and flavor, as well as their nutraceutical and pharmacological properties, playing a fundamental role in defence against the diverse stresses faced by these plants [50]. An increase in GSL content in broccoli plants was described when plants were grown under salinity for 15 days [4]. After measuring their GSL content, a marked increase was observed in the leaves of plants treated with 80 mM NaCl compared with the control. These results could be associated with new expression or activation of aquaporins resulting in increased water permeability in tissue [4]. This finding suggested a role played by GLs in the plant’s water response under salinity. Furthermore, Keiling and Zhujun [51] reported an increase in GSL levels in pak choi (*Brassica campestris* L. ssp. *chinensis* var. *communis*) after plants were subjected to salt stress. Plants treated with 50 mM NaCl increased their levels of aliphatic and indole GSLs, while those treated with 100 mM NaCl considerably increased their levels of indole GSLs and decreased their levels of aromatic GSLs. These results show how the profile of GSLs in these plants is strongly influenced by salt stress, indicating their key role as signaling molecules for the regulation of stomatal opening and closing, or water transport through plants [52], and their potential effect on auxin signaling [53].

### 3.2. Drought

The sensitivity of plants to drought and heat stress has been mainly associated with accelerated production of reactive oxygen species (ROS), which causes strong oxidative stress [54]. Therefore, plants have developed different defence systems to avoid this damage, such as the overproduction of antioxidant metabolites capable of stopping the propagation of oxidative chain reactions. These polyphenolic compounds include phenolic acids, flavonoids, proanthocyanidins, and anthocyanins [55]. Sarker et al. (2020) [56] found an association between a high content of phenolics and the high drought resistance of certain amaranth vegetables, with the most abundant compounds being flavonoids, benzoic acids, and cinnamic acids. An increase in the production of phenolic compounds in response to moderate drought has been reported in plant species such as *Salvia officinalis* L., with their production capacity found to be lower under severe stress [57]. Together these findings suggest that the ability of plants to combat oxidative stress, and therefore drought, significantly depends on their capacity to modulate the synthesis of phenols. In addition, *E. urophylla* grown under water deficit increased its concentrations of wall-bound phenolics, as a strategy for increasing water transport and improving the use of water resources [58]. Thus, it has been reported that phenolics such as salicylic acid regulate aquaporins for increasing the hydraulic properties of roots in arbuscular mycorrhizal maize plants subjected to drought [59].

In relation to other secondary metabolites such as GSLs, it has been suggested that these could possibly have a role in osmotic adjustment, given the changes in their composition that plants exhibit during drought [4,60]. Several studies agreed that aliphatic GSL content was increased under water stress [61]. It was shown that *Arabidopsis thaliana* plants subjected to drought were able to withstand these conditions, due to an increase in water uptake related to the increase in the concentrations of aliphatic GSLs occurring through the activation of a transcriptional cascade mediated by the auxin-sensitive repressor Aux-IAA [62]. The loss of this repressor triggered a reduction in the aliphatic GSL content, and therefore in their tolerance to drought, highlighting the importance of these GSLs in drought tolerance.

Meanwhile, research on *A. thaliana* and Chinese cabbage showed that the drought-induced accumulation of GSLs in the leaves of these plants directly or indirectly controlled the closure of stomata, thus preventing water loss [38,62]. Furthermore, the increase in GSLs in canola grown under water stress was related to the steady water budget and the high accumulation of proline that protected the photosynthetic pigments [63].

### 3.3. Heavy Metals

Soil pollution caused by heavy metals is one of the major problems currently faced in agriculture, and has adverse effects on food safety. Despite the fact that some metals are essential micronutrients in plants, their excess can have adverse effects on plants’ growth, metabolism, physiology, and senescence, as a consequence of ROS accumulation [64].

Phenols, as antioxidant compounds, have the ability to chelate metals depending on their hydroxyl group, especially copper (Cu) and iron (Fe). In addition, they bind to ROS such us hydrogen peroxide and superoxide ions, thus reducing oxidative stress [65]. Furthermore, soluble phenolic compounds such as lignin-synthesis intermediates enhance the strength of the cell wall, thus creating physical barriers that alter elastic modulus for maintaining water uptake to protect the cells [66]. Due to these facts, it is not surprising that several studies reported increases in phenolic total contents related to the tolerance of the plants when heavy metals such as cadmium (Cd), zinc (Zn), Cu, or lead (Pb) were present [66,67].

The Brassicaceae family is characterized by a large number of species that are hyper-accumulators of heavy metals, in particular nickel (genera Thlaspi and Alyssum), cadmium, and zinc (*Thlaspi caerulescens, Thlaspi praecox, Thlaspi goesingense*, and *Arabidopsis halleri*) [68]. They present cellular mechanisms of resistance including compartmentalization in relation to the increase of GSL levels. Furthermore, an increase in the content of GSLs, especially indolic GSLs, was observed in the roots of Chinese cabbage exposed to Cu [69] and enhanced biosynthesis of GSLs was observed in white cabbage exposed to Cd and Zn [70]. It has been suggested that such increases in GSL contents might be related to water management in leaves, providing higher protection against stress.

## 4. Movement of Phenolic Compounds and Glucosinolates in Plant Root Exudation

Secondary metabolites, including GSLs and phenolic compounds, are distributed bidirectionally between leaves and roots [71]. In this distribution of compounds, intracellular movement occurs via membrane transporters through the ABC (ATP binding cassette), MATE (multidrug and toxic extrusion compound), and NRT/PTR (nitrate/peptide transporter) families [72,73], or via vesicle-mediated transport [74] (Figure 3). Although current knowledge is limited, advances have been made in elucidating the molecular mechanism of intracellular trafficking and the exudation mechanism of these secondary metabolites.

When in the roots, these compounds can be secreted or exuded into the surrounding soil, also called the rhizosphere. As the root system not only plays the role of an anchor but also functions for the uptake of nutrients and water, its key role in plant–soil–plant communication is being studied [75]. The plant’s interactions with nearby plants and microbes is related to the exudation of a wide variety of compounds such as phenolics and GSLs. The importance of root exudation is evident, as plants invest up to 20% of photosynthetically fixed carbon in root exudates [76].

Root exudates can be delivered through a passive or active process [77]. The passive processes include diffusion, which involves electrochemically positive concentration gradients between the cytoplasm of root cells and the external solution or soil, ion channels, vesicle transport, and aquaporins. Active secretion involves specific membrane-bound transport proteins located in the plasma membrane, such as those mentioned above, ABC or MATE transporters, or the aluminum-activated malate transporter (ALMT) which is part of the major facilitator superfamily (MFS). Most of these exudation processes (passive or active) occur at the root tip, which is also the first part of the plant to explore the changing soil conditions [75,78]. Therefore, root exudation has a key function in the response to environmental changes. It has been described that the quantity and composition of root exudates depend not only on the plant species, stage of development, nutrition, and soil type, but also on external factors such as abiotic and biotic stressors (nutrient and/or water deficits, pathogen and pest attacks, or heavy metal contamination) [79,80]. Root exudates contain inorganic acids, oxygen, ions, water, and organic compounds (amino acids, organic acids, sugars, phenolics, GSLs, or proteins) [81]. In this context, alterations in the composition of root exudates in response to stress is continuously being studied [82,83].

Several assays have shown modifications in the profiles of phenolic compounds in root exudates in response to abiotic stresses. Generally, an increase in root exudates of phenolic compounds has been identified as a response to mineral nutrient stress, contributing to acquisition of mineral nutrients by plants through the modification of the rhizosphere, to enable access to low levels of nutrients available or to avoid toxicity [84]. In this way, aluminum (Al) toxicity, one of the most limiting factors for plant growth, triggers the release of different phenolic compounds into the rhizosphere. For example, catechol, catechin, quercetin, and curcumin were exuded by Al-exposed *Zea mays* [85]. That study proposed these phenolic compounds as important new elements of root exudates associated with the detoxification of Al in the root tips of maize. This role had previously been attributed to organic acids such as malate or citrate. Recently, the role of phenolics in alleviating metal toxicity was shown to be of interest when dealing with certain endangered species such as *Kandelia obovate*, which is frequently subjected to excess metals in its environment. Exudation of phenolic acids with strong antioxidant ability was observed in this type of plant [86]. Copper (Cu) toxicity also triggers the release of phenolic compounds by the roots [87,88,89]. Different responses were found according to plant type and tolerance against these types of stress (metallophytes or agricultural plants). Catechin, cinnamic acid, and coumaric acid were detected in the root exudates from different plants, and results were dependent on Cu concentration and time [89]. From the results of these studies, it is possible to conclude that phenolic compounds act as an efficient exclusion mechanism to prevent Cu uptake and subsequent toxicity. Nutrient deficiencies such as iron (Fe), phosphorus (P), or zinc (Zn) deficiency have also been reported to modify the phenolic compound profile in root exudates. For example, increases in phenolic compounds in root exudates were reported for different species grown under conditions of Zn deficiency [90]. In this regard, the exudation into the rhizosphere of two coumarin-derived compounds, sideretin and fraxetin, was induced under Fe deficiency in *Arabidopsis thaliana*. When in the rhizosphere, these redox-active coumarins solubilize and reduce Fe from scarcely available sources [83]. This response was reportedly much more intense when the Fe accessibility was lower or Fe status was deteriorated, for example, in a growth medium with a more basic pH [91]. In the root exudates of different eucalyptus species subjected to low P concentrations, compared with eucalypts under sufficient P, higher total amounts of phenolic compounds were found, with cinnamic acid prominent among these [92].

Aspects of water stress have been widely studied in all aspects, as it is one of the most limiting factors for plant growth and productivity. Different tests have been carried out on phenolic compounds released into the rhizosphere. Drought-altered root-exudate composition was assessed in terms of phenolic acids, lignans, flavonoids, and flavones in two genotypes (drought-stress sensitive and tolerant) of *Pennisetum glaucum* (pearl millet) [93]. Surprisingly, flavonoids were not identified under well-watered conditions (control) but were detected under drought stress [93], which demonstrates how the exudation profile of the roots can change depending on whether the plants grow under water stress. This increase in flavonoids is probably related to the fact that these compounds act as antioxidants and play a key role in defence and signaling mechanisms [94]. A study conducted under salt stress (50 mM) revealed that the pattern of flavonoids detected in root exudates from *Phaseolus vulgaris* cv. Bush Blue Lake did not change, although quantification of different flavonoids under different conditions was not attempted, with only basic identification carried out [95]. Therefore, the study results do not allow us to determine whether salinity caused changes in the concentrations of flavonoids present in the exudates of the plant.

Regarding GSLs, there has been little study whether or not the GSL profile in root exudates is altered under stress. For example, elicitation with methyl jasmonate increased the concentration of indole GSLs in root exudates of *Brassica rapa* ssp. *Rapa* (turnip) [96]. Methyl jasmonate (MeJa) is not an environmental stress per se, but is considered a biotic stress. MeJa causes the induction of reactive oxygen species (ROS) and the triggering of defence mechanisms [97], including the production, accumulation, and exudation of GSLs by the roots. Likewise, Rios et al. (2021) [98] in a study on broccoli showed that MeJa applied by foliar fertilization triggered an increase in the concentrations of different GSLs and isothiocyanates (ITCs) in root exudates. GLS such as methoxyglucobrassicin (MGB) cinnamoyl (feruloyl)-indol-GLS (F-GSL), and ITCs such as sulforaphane (SFN) and indole-3-carbinol (I3C) increased two-fold. In addition, broccoli root exudates under salinity stress were also analyzed, and that stress led to similar results in terms of the amounts of GSLs and ITCs present in the exudates, with increases in these compounds also shown [98]. The presence of GSLs and ITCs in root exudates may be related to a defence mechanism against biotic stresses such as pathogens, as these compounds have been demonstrated to exert antimicrobial activities [50,99]. However, the role of these compounds under salinity stress or other water stresses has not been fully explained. GSLs and their hydrolysis products (ITCs) have been shown to alleviate the impact of abiotic stresses such as salinity, drought, or high temperature, but the molecular mechanism of this alleviation process has scarcely been investigated (Table 1).

## 5. Cells Transporters of Phenolic Compounds, Glucosinolates and Aquaporins

Compounds such as secondary metabolites are metabolically costly to produce and, according to the optimal defense theory, tend to accumulate at higher levels in tissues that are sensitive or more likely to be attacked [100,101]. Therefore, transport processes are essential to achieve energetic optimization of plant biology through the reallocation of these metabolites to the target tissues, as exemplified by a decrease in these metabolites in senescing vegetative tissues accompanied by their increase in reproductive tissues.

Long-distance GSL transporters have been studied for only a few years, due to their recent discovery in *Arabidopsis* [72]. In particular, two specific GSL importers, GTR1 (NPF2.10) and GTR2 (NPF2.11), both members of the nitrate/peptide transporter family, have been described as essential for the long-distance transport of GSLs from their source tissues to target organs such as seeds or young leaves. It has been hypothesized that this transport probably occurs via phloem companion cells [72]. The two GTR transporters have been localized to the plasma membrane, and both function as high-affinity H^+^/GSLs symporters that appear to be pH-dependent, suggesting a role in the importation of GSLs into the cytosol from the apoplasm or even the vacuole. Of the two GTR transporters identified, GTR2 has been hypothesized to play an important role in the loading of GSLs into the apoplasm, as its lack led to a decrease in GSL content in Arabidopsis seeds [72]. In contrast, GTR1 has been linked to the distribution of GSLs within the leaf, possibly involved in importation into the GSL-rich-S-cells located adjacent to the phloem. It has been shown that BocGTR1a and BocGTR1c, both isoforms of GTR1 present in Chinese cabbage, may be involved in GSL accumulation in different plant tissues [102]. In this scenario, the influence of aquaporins on the long-range transport of GSLs is evident, as the water status of plants is related in a non-direct manner to the activity of aquaporins and motivates the transport of metabolites via the xylem and phloem.

However, phenolic compounds are a very heterogeneous group, encompassing phenolic acids, flavonoids, tannins, stilbenes and lignans, and a specific transporter for all of them has not yet been identified. Indeed, one is not expected to be found because it is impossible to have a specific transporter for each phenolic compound or type. Little is known about the transport and mobilization processes of these metabolites in plants. In recent years, several hypotheses have been put forward to try to unravel the transport of phenolic compounds within the cell, i.e., movement towards the apoplast and vacuolar sequestration [103], from the involvement of vesicle trafficking to membrane transporters or even mediation by glutathione S-transferase (GST). Among all the possibilities, it is thought that the transport of these metabolites is carried out by a mixture of these three non-exclusive mechanisms functioning in an integrated manner for vacuolar storage and apoplast secretion pathways. Focusing on the pathway involving transmembrane transporters, two families have been described as possible phenolic transporters, specifically flavonoid, anthocyanin, and isoflavonoid transporters. The ATP-binding cassette (ABC) transporter family consists of transmembrane proteins involved in the transport of a wide range of different molecules such as hormones as well as primary and secondary metabolites [104]. Different members of this family have been linked to anthocyanin accumulation, such as ZmABCC3,ZmABCC4, MRP3, and MRP 4, and EC 7.6.2.2 which presumably transports glutathione-conjugated flavonoids into the vacuole in maize [105]. Members of the multidrug and toxic compound extrusion (MATE) transporter family have also been found to be involved in the transport processes of phenols such as flavonoids, located in the tonoplast, facilitating vacuolar sequestration [106]. For instance, numerous MATE transporters have been associated with the transport of phenolics, such as FaTT12-1 in the accumulation of proanthocyanidin in strawberry fruits [107], or GmMATE1 which is reported to promote the accumulation of isoflavones in soybean seeds [108].

MATE transporters have also recently been characterized as voltage-dependent chloride anion channels, facilitating chloride entry into the vacuole and regulating turgor pressure and cell expansion, and acting as regulators in important processes such as stomatal movements in guard cells [109]. In this sense, MATE transporters and aquaporins should work in a coordinated manner, as stomatal opening and closing are finely regulated with the involvement of aquaporins, MATE transporters, and other agents. Although the biosynthesis of phenolic compounds in plants has been well described, little is known about the transport processes of these metabolites within the cell, with intercellular and long-distance transport among the most poorly understood aspects of their activity. Consequently, efforts should focus on deciphering these processes to achieve a complete understanding of plants’ secondary metabolism, transport, and regulation under abiotic stress response.

## 6. Effect of the External Application of Phenolic Compounds and Glucosinolates

As described throughout this review, phenolic acids and GSLs are both involved in numerous resistance strategies to abiotic stresses [110,111,112,113], which is why certain current trends involve their foliar application to obtain these benefits. One of the phenolic acids most frequently utilized as a foliar treatment is salicylic acid (SA). SA is a phenolic compound that regulates antioxidant activity, and is also an endogenous signaling molecule that induces stress tolerance against abiotic stresses (drought, heavy metal, and salt stress) [111]. SA has been proven to ameliorate the negative effects of salt and water deficit stresses, acting as an antioxidant, up-regulating the synthesis of photosynthetic pigments, improving leaf water status, and enhancing antioxidant systems [110,111,114].

In cinnamon plants (*Ocimum basilicum* L.), SA application at different concentrations (from 0.29 mM to 2 mM) attenuated the harmful effects of irrigation with salty water, and increased gas exchange and total chlorophyll [110]. In *Torreya grandis* L. trees grown in 0.2 and 0.4% NaCl conditions, the foliar application of SA effectively increased the chlorophyll content, relative water content (RWC), net CO_2_ assimilation rate (Pn), and proline content. In addition, it enhanced the activities of superoxide dismutase (SOD) and peroxidase (POD), and minimized the increases in relative leakage conductivity (REC) and malondialdehyde (MDA) content [114]. In barley plants (*Hordeum vulgare* L. cv. Gustoe), the addition of 50 µM SA decreased oxidative stress, the Na/K ratio, and MDA content [115]. SA provided through the roots was also able to produce a similar positive effect against salinity stress in chili (*Capsicum annuum* L. cv. Pusa Sadabahar) [111].

Some studies have indicated that endogenous SA plays a fundamental role in the early stages of nodulation in Rhizobium-legume symbiosis [116], and others revealed that the exogenous application of 100 µM SA inhibited indeterminate nodulation of *Rhizobium leguminosarum* L. bv. *viciae* [117] and the same concentration inhibited early nodulation in soybean plants [118]. Therefore, it is not surprising that the external application of SA favors this process. Foliar application of 10 mM SA was shown to increase plant height, and the application of 1 mM SA showed the highest increase in nodules per plant [116]. In this way, numerous phenolic acids act as signal molecules, establishing arbuscular mycorrhizal symbiosis, as is the case with hydroxybenzoic and hydroxycinnamic acids [14]. Other phenolic acids have shown to interact with the nodules formed, either influencing the growth of rhizobia, such as the case of *p*-coumaric acid [119], or forming part of the structure, as with caffeic acid [120]. This increase in nodules could point to higher N_2_ fixing, allowing greater plant growth, but it could also be involved in an increase in water permeability through aquaporins, especially NIPs (nodulin 26-like proteins) [121], further improving growth under conditions of water stress [122].

Other foliar treatments with phenolic acids, such as ferulic acid (10–100 µM), have also been demonstrated effective against abiotic stresses. For example, Chinese cabbage (*Brassica rapa* L. ssp. *pekinensis*) that underwent short-term (72 h) salinity stress (150 mM NaCl) showed an increased number of phenolic compounds and an improved rate of photosynthesis [123]. This same phenolic acid applied to pea plants (*Pisum sativum* L.) at 100 and 150 ppm induced phenylalanine ammonia-lyase (PAL) activity, an enzyme that is involved in defense responses [124]. Other complex mixtures, composed partly of phenolic compounds, have also been used in foliar applications, such as algae extracts (*Spirulina platensis* L., *Chlorella vulgaris* L., *Amphora coffeaeformis* L. and *Scenedesmus obliquus* L.) used on cardoon plants (*Cynara cardunculus* L.). The application of these extracts increased the size of the plants and their accumulation of oils and carbohydrates. The phenolic acids that in higher concentrations included chlorogenic, caffeic, and vanillic acids [125]. Others act as antioxidants, such as gallic acid and its methyl ester [126], and can also be produced in response to water stress [127]. However, the foliar application of these phenolic acids requires further investigation.

Other important phenolic compounds are flavonoids and their derivatives. Some studies have related their foliar application to a better stress response, as flavonoids diminish the activity of superoxide anions and reduce oxidative stress [128]. For example, 16.67 mg to 166.67 mg of rutin, a derivate of flavonoids, was sprayed onto *Amaranthus hypochondriacus* L. The addition of rutin caused significant effects on the synthesis of glutathione (GSH), including promotion of the conversion of GSH to phytochelatins (PCs), contributing to a higher accumulation of Cd, with improvements up to 219.48% and 260% [129]. Furthermore, the report described an alteration of the permeability of cell membranes, with an increase in the immobilization of Cd ions in the vesicles, which allowed the cells to tolerate stronger Cd stress.

Foliar application of vanillic acid, an intermediate of ferulic acid, and p-hydroxybenzoic acid, enhanced drought tolerance and the formation of phytoalexin momilactones in rice, while total phenolics, flavonoids, pigments, and antioxidant activity were also positively promoted [130]. Metabolic pathways of phenolic compounds are closely associated, as the application of one influences and alters the quantities of others. For example, SA application modifies and interacts with other phenolic acid pathways, and the foliar application of salicylic acid was proven to increase caffeic acid derivatives in all plant organs in purple coneflower (*Echinacea purpurea* L.) [131]. Furthermore, concentrations ranging from 0.15 to 0.50 mM of SA can increase the concentrations of total phenolic and flavonoid compounds as well as fruit yield in cucumber (*Cucumis sativus* L.) [132]. In addition, concentrations ranging from 0.075 to 0.15 mM increased the firmness of the fruit, and these fruits had less water loss, thereby improving the quality of the fruit for a longer time, and signaling the possible regulation of aquaporins by these compounds.

### Foliar Application of Glucosinolates

The foliar application of GSLs is beginning to be used, due to the many benefits it brings. The foliar application of GSLs and their derivatives such as isothiocyanates has been described as an effective tool against parasitic nematodes [133,134], with their toxicity to nematodes utilized to control associated biotic stresses. Insecticidal properties have been proposed for GSLs, and they have been demonstrated to increase the mortality of aphids [135]. On other hand, other GSLs such as sinigrin have shown effectiveness against abiotic stresses including salt stress [136], increasing hydraulic conductance and water permeability, and raising the concentration of PIP2 in *Brassica oleracea* L. plants. GSLs have also been shown to affect and improve the yield of some plants. For example, different foliar applications of GSLs (GSL with mineral oil as coadjuvant, GSL-based formulate, GSL–water suspension) increased tomato yields by improving plant health, with the highest increase observed with the combination of GSL and mineral oil, as indicated by an increase in yield of up to 28% [137], possibly due to improved absorption of GSLs. However, the mechanisms by which GSL improves crop yield have not been described, opening new lines of research and pointing to the increase in aquaporins and greater water availability as possible associated factors.

## 7. Concluding Remarks

The biosynthetic pathways for secondary metabolite phenolic compounds and GSLs have been highly studied in plants, but their intervention in responses to abiotic stress has been poorly documented. As we observed in all the reports, the concentrations of these compounds increased when plants were subjected to abiotic stresses, pointing to their role in mitigating the effects of stress. However, individual chemical species may have specific roles that need further investigation and could serve for the design of biostimulants for external applications. In this context, the described capability of phenolic compounds and GSLs as ROS scavenging agents may be related to an improvement of water uptake and transport. The fact that these metabolites are part of the root exudates, with high associated energy costs, suggests their adaptive effect for improving nutrient remobilization and water uptake under abiotic stress. However, the molecular mechanisms underlining this response need to be investigated. Investigations into the processes determining their synthesis, transport, and exudation should be completed, including assessment of the regulation of membrane transporters.

## Figures and Tables

**Figure 1 ijms-24-02826-f001:**
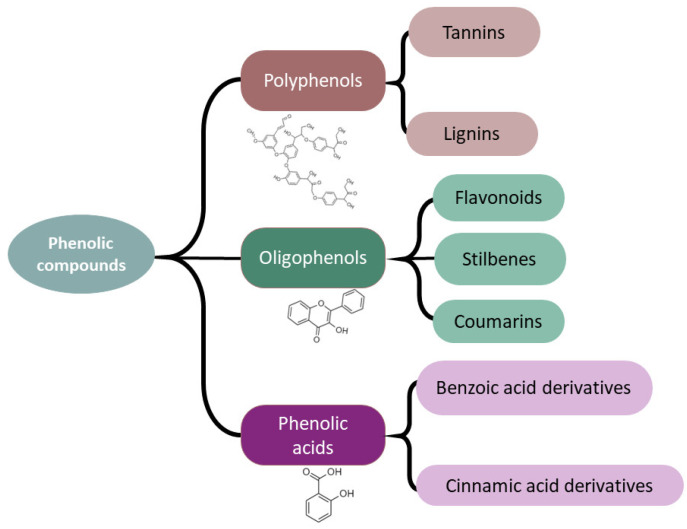
Scheme of the main types of phenolic compounds present in vegetables.

**Figure 2 ijms-24-02826-f002:**
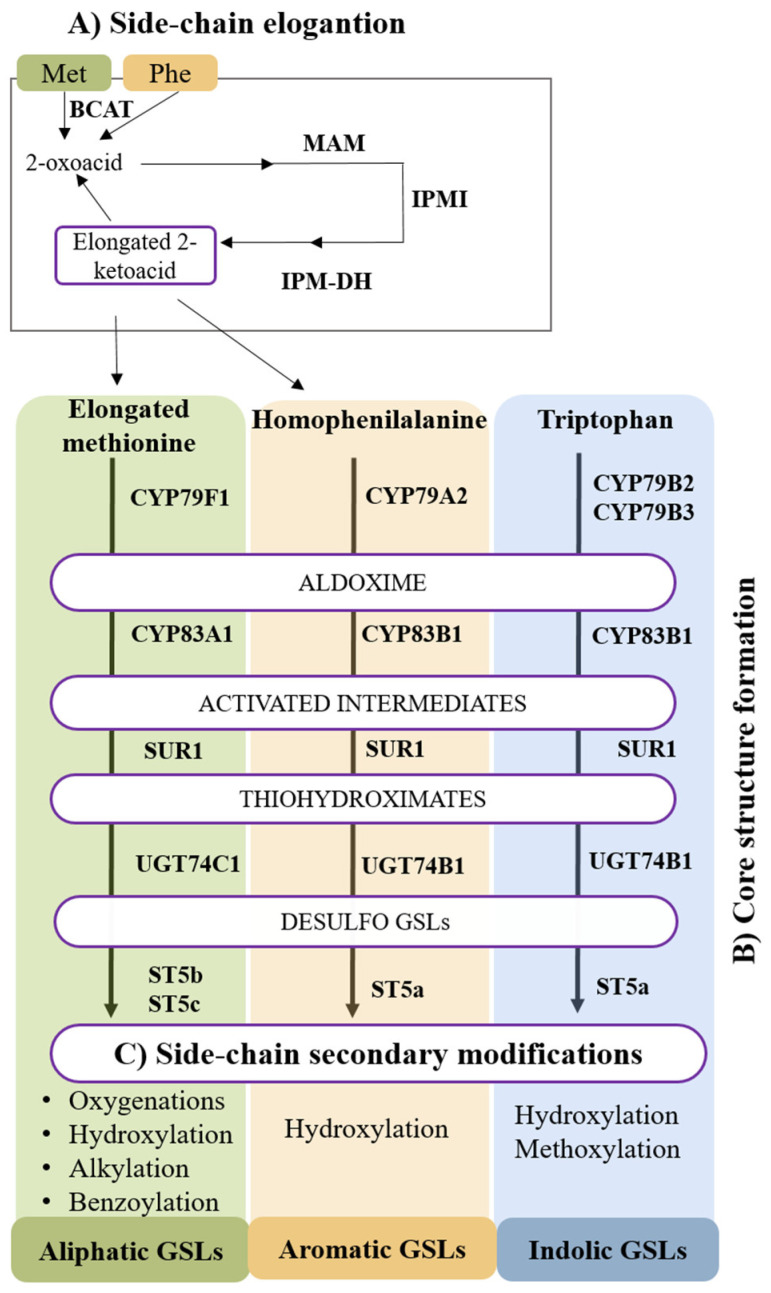
Biosynthetic pathways of alphatic, aromatic, and indolic glucosinolates. (**A**) Side-chain elongation: BCAT, branched-chain aminoacid transferase; MAM, methylthioalkylmalate synthase; IPMI, isopropylmalate isomerase; IPMDH, isopropylmalate deshidrogenase. (**B**) Core structure formation: CYP79F1, CYP79A, CYP79B2, and CYP79B3, cytrochromes P450 from the family CYP79; CYP83A1 and CYP83B1, cytochromes P450 from the family CYP83; SUR1, super root 1; UGT74C1 and UGT74B1, UDP-glycosyltranferse; ST5, sulphotransferase 5. (**C**) Secondary side-chain modifications.

**Figure 3 ijms-24-02826-f003:**
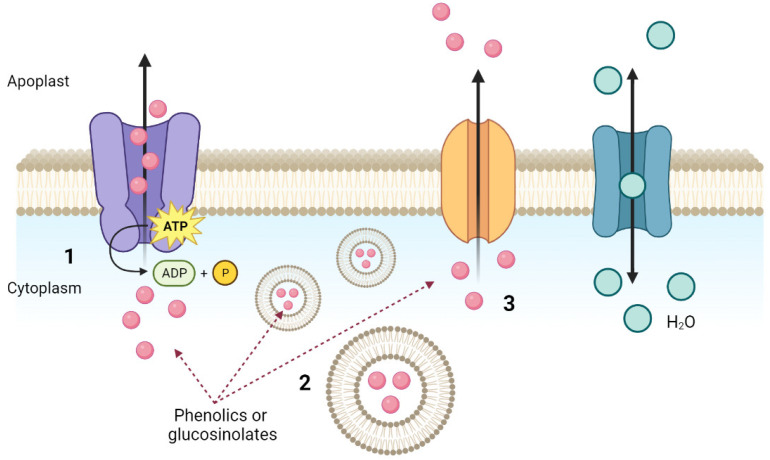
Mechanisms of cell excretion of phenolic compounds and glucosinolates by root through membrane. (1) The active process; (2) vesicle trafficking; (3) the diffusion channels. In all cases the involvement of aquaporins is necessary to facilitate the process.

**Table 1 ijms-24-02826-t001:** Summary of the GSLs and phenolic compounds exuded by the roots in the rhizosphere under different stress conditions.

Compound Name	Plant Species	Treatment/Growing Condition	Modification	Ref.
**Phenolic Compounds**				
Naringenin, isoiquiritigenin, quercetin umbelliferone, 7′,4-dihydroxyflavone, hesperetin	*Phaseolus vulgaris*	Salinity	Same profile. *no quantification	[95]
Flavonoids	*Pennisetum glaucum*	Drought	**↑**	[93]
Sideretin and fraxetin	*Arabidopsis thaliana*	Fe deficiency	**↑**	[83]
Cleomiscosins, 5′-hydroxycleomiscosins, scopoletin, fraxetin, isofraxidin, and fraxinol	*Arabidopsis thaliana*	Fe deficiency (pH 7.5)	**↑**	[91]
Cinnamic acid	*Eucalyptus*	P deficiency	**↑**	[92]
Phenolics (caffeic acid equivalent)	*Malus domestica, Phaseolus vulgaris, Triticum aestivum*	Zn deficiency	**↑**	[90]
Protocatechuic acid, ferulic acid, and cinnamic acid	*Kandelia obovata*	Cd and Zn toxicity	**↑**	[86]
Catechin	*Imperata condensate*	Cu toxicity	**↓**	[89]
Cinnamic acid	*Oenothera picensis, Imperata condensate, Lupinus albus, and Helianthus annuus*	Cu toxicity	=	[89]
Coumaric acid	*Helianthus annuus*	Cu toxicity	**↑**	[89]
Catechin, catechol, curcumin, and quercetin	*Zea mays*	Al toxicity	**↑**	[85]
**Glucosinolates**				
2-phenylethyl (gluconasturtiin)	*Brassica rapa *spp.* rapa*	Salicylic acid and methyl jasmonate addition to the nutrient solution	**↑**	[96]
Methoxyglucobrassicin, Cinnamoyl (feruloyl)-indol-GLS	*Brassica oleracea var. italica*	Methyl jasmonate foliar elicitation and salinity	**↑**	[98]

## Data Availability

Not applicable.

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
