# Peer review of "Confronting Secondary Metabolites with Water Uptake and Transport in Plants under Abiotic Stress"

_ijms, 2023, doi:10.3390/ijms24032826_

Round 1

Reviewer 1 Report

It is a good attempt by the authors to highlight the potential role of secondary metabolites in plant adaptation to various environmental stresses. However, some significant revisions will be needed and the suggestions are mentioned below:

1. Title

The usage of secondary metabolism sounds inappropriate, as the review focuses on secondary metabolites and their role in abiotic stress tolerance. So, the title may be revised to a suitable one, clearly reflecting the content of the review.

2. Abstract

The abstract needs to be rewritten as the language is not proper (e.g., line 11: this review will evaluate; line 12: role of phenolics and glucosinolates in the interactions of plants with abiotic stresses; water uptake and transport mediated through aquaporins was reviewed in line 15). It is not clear what the authors want to convey from the last sentence.

3. Main text

1.  Section 2

Physiological role of phenolic compounds presented in the subsection 2.2 can be little more elaborated citing some more references reflecting a synthesis of previous reports and recent advances in plants secondary metabolites with reference to abiotic stresses in crop plants. In line 116, it is written as “Lignin is involved in important vegetative roles in plant growth”, the word vegetative role seems improper, so it may be changed accordingly.

2. Section 3

As mentioned in the title itself, the major aim of the review is to present a details analysis of how phenolic compound would play a role in improving the water uptake and transport under abiotic stress environment. Although few references have been cited in this line of work, however, the content is inadequate given the title, which appears to be more like reporting of the phenolic compound induction in response to abiotic stress. It is suggested to present this section with more focus on the potential role of phenolic compounds relating to plant water status improvement substantiated by appropriate references.

Typological errors:

i.                 line 114: cell structural roles is typed in bold

ii.                line 183: their mayor bioactivity relies on their degradation products

iii.               line 291: Phenolic compounds and glucosinolates movement in plant. Root exudation

iv.               line 565: specie

Author Response

It is a good attempt by the authors to highlight the potential role of secondary metabolites in plant adaptation to various environmental stresses. However, some significant revisions will be needed and the suggestions are mentioned below:

  1. Title

The usage of secondary metabolism sounds inappropriate, as the review focuses on secondary metabolites and their role in abiotic stress tolerance. So, the title may be revised to a suitable one, clearly reflecting the content of the review.

R: The title has been changed as suggested the referee

  1. Abstract

The abstract needs to be rewritten as the language is not proper (e.g., line 11: this review will evaluate; line 12: role of phenolics and glucosinolates in the interactions of plants with abiotic stresses; water uptake and transport mediated through aquaporins was reviewed in line 15). It is not clear what the authors want to convey from the last sentence.

R: The sentences have been clarified and rewritten.

  1. Main text
  2.  Section 2

Physiological role of phenolic compounds presented in the subsection 2.2 can be little more elaborated citing some more references reflecting a synthesis of previous reports and recent advances in plants secondary metabolites with reference to abiotic stresses in crop plants. In line 116, it is written as “Lignin is involved in important vegetative roles in plant growth”, the word vegetative role seems improper, so it may be changed accordingly.

R: Thank for the appreciation, further knowledge in the physiological role related with abiotic stress has been included in lines 115 to 122. Also, the mentioned sentence has been rewritten. 

  1. Section 3

As mentioned in the title itself, the major aim of the review is to present a details analysis of how phenolic compound would play a role in improving the water uptake and transport under abiotic stress environment. Although few references have been cited in this line of work, however, the content is inadequate given the title, which appears to be more like reporting of the phenolic compound induction in response to abiotic stress. It is suggested to present this section with more focus on the potential role of phenolic compounds relating to plant water status improvement substantiated by appropriate references.

Typological errors:

-line 114: cell structural roles is typed in bold

R: The typological error has been corrected in the manuscript.

-line 183: their mayor bioactivity relies on their degradation products.

R: The verb has been corrected.

- line 291: Phenolic compounds and glucosinolates movement in plant. Root exudation

R:The sentence has been rewritten in the manuscript.

-line 565: specie

R: “Specie” has been changed to “species” in the manuscript.

Reviewer 2 Report

The current review article entitled “Confronting secondary metabolism with water uptake and transport in plants under abiotic stress” by Nicolas-Espinosa et al. is a very comprehensive and interesting idea however the manuscript needs to be improved.

1. The focus is on phenolic compounds only while title mentions all secondary metabolites. Polyphenols should also be discussed which fall under the category of phenolic compounds and are involved in abiotic stress response as well.

2. Salinity, drought, and high temperature stresses are discussed. Phenolic compounds also have pronounced effect under drought, heavy metal and nutrient deficiency stresses which should also be discussed.

3. The manuscript can be improved by avoiding excessive use of indirect language especially in the “1. Introduction” and “7. Concluding” remarks section. 

4. The manuscript lacks visual representation. Including more figures will greatly improve the review.

5. Heading 2.1, 2,3; Schematic diagram of phenolic compound and glucosinolate biosynthesis would have been quite helpful in understanding.

6.      Line 149-153; abbreviations are used without mentioning the full form.

7.      Line 303-304; The figure caption needs grammatical correction.

8.      Concluding remarks are poorly written. 

Author Response

The current review article entitled “Confronting secondary metabolism with water uptake and transport in plants under abiotic stress” by Nicolas-Espinosa et al. is a very comprehensive and interesting idea however the manuscript needs to be improved.

  1. The focus is on phenolic compounds only while title mentions all secondary metabolites. Polyphenols should also be discussed which fall under the category of phenolic compounds and are involved in abiotic stress response as well.

Pholyphenols has been included and discussed as beneficial agents against abiotic stresses in lines 127 to 133.

  1. Salinity, drought, and high temperature stresses are discussed. Phenolic compounds also have pronounced effect under drought, heavy metal and nutrient deficiency stresses which should also be discussed.

Further information against heavy metal, drought and nutrient deficiency has been included in lines 117 to 123.

  1. The manuscript can be improved by avoiding excessive use of indirect language especially in the “1. Introduction” and “7. Concluding” remarks section. 

The whole manuscript has been reviewed accordingly, mainly introduction and conclusion

  1. The manuscript lacks visual representation. Including more figures will greatly improve the review.

Some figures have been added

  1. Heading 2.1, 2,3; Schematic diagram of phenolic compound and glucosinolate biosynthesis would have been quite helpful in understanding.

The schematic diagram has been added in both sections.

  1. Line 149-153; abbreviations are used without mentioning the full form.

“Quantitative Trait Locus” referring to "QTLs" has been added.

7.Line 303-304; The figure caption needs grammatical correction.

It has been corrected.

Round 2

Reviewer 1 Report

All the comments have been satisfactorily addressed. No further comments.

Reviewer 2 Report

The authors have significantly improved their article. I think now it is suitable to be considered for publication in IJMS.